# Effect of Rill Development on Slope Erosion and Sediment Yield Based on Stereophotogrammetry Technology

**Zilin Zuo [1,2], Hui Wang [3,*], Shuwen Ding [4] and Yijin Wu [1,*]**

1 School of Urban and Environment Science, Central China Normal University, Wuhan 430079, China
2 Network and Information Center, Changjiang River Water Resources Commission, Wuhan 430010, China
3 Hubei Anyuan Safety and Environmental Protection Technology Co., Ltd., Wuhan 430000, China
4 College of Resources and Environment, Huazhong Agricultural University, Wuhan 430070, China
* Correspondence: wanghui001001@163.com (H.W.); yijinwu202208@163.com (Y.W.)

**Abstract:** Rill erosion is an important kind of slope erosion and the main source of sediment. Through simulated rainfall tests, the morphological characteristics of rill were quantified by stereophotogrammetry technology, and the relationship between rill development and sediment yield was studied. The results show that there was a positive correlation between sediment yield and slope and rainfall intensities. With the increase in rainfall duration, sediment yield first increased sharply and then decreased gradually after reaching the peak value, until it reached dynamic stability. With the increase in rainfall intensity and slope, the length, width, and number of rills increased significantly, with a maximum length of 2.58 m and a maximum width and depth of 9.7 and 2.2 cm. The rill density (RD) increased from 16.67% to 62.65%; rill fragmentation degree (RFD) increased from 16.67% to 100.00%; rill complexity (RC) increased from 10.62% to 30.84%, and rill width–depth ratio (RWDR) decreased from 15.82% to 56.28% with the increase in slope from 6° to 15° and rainfall intensity from 2.0 to 3.0 mm/min. There was a good nonlinear relationship between sediment yield and RC and RWDR ($R^2$ = 0.89, NSE = 0.85, n = 10). This study could provide help for the quantification research of rill erosion mechanisms and provide reference for the measurement and scale transformations of soil erosion at different scales.

**Keywords:** soil; rill; stereophotogrammetry; erosion; sediment yield





## 1. Introduction

Soil erosion has become one of the most critical environmental problems influencing sustainable development and agricultural productive capacity [1,2]. Rill erosion is an important mode of erosion, which is between surface erosion and gully erosion, which is formed and developed when the erosivity of rainfall and runoff is greater than the erosivity of soil [3]. Once rill forms on a slope, the slope flow will converge into concentrated rill flow by inter-rill flow, and the slope runoff velocity and flow depth will increase to varying degrees, leading to the intensification of slope soil erosion [4–6]. Studies have shown that rill erosion can account for more than 90% of total slope erosion [7,8]. Studies have been conducted on rill erosion quantity [9], rill erosion form [10,11], and rill erosion development process [12,13], and a series of progress has been made.

Rill erosion and sediment yield occurred during the development of rill erosion, so rill morphology changed rapidly, and the rill response to change will react on slope erosion and sediment yield. Rill development has three stages: formation, development, and adjustment. Many factors (rainfall intensity, slope, soil grain composition, etc.) affecting rill erosion were studied by experiments [3,14,15]. The inflow rate exerted the greatest effect on the average rill width and average rill depth, with contribution rates of 73.11% and 60.26%, respectively. Meanwhile, slope exerted the greatest effect on the rill width-to-depth ratio, with a contribution of 60.45% [16]. Rill length has a linear positive correlation with time, and sediment yield was affected by the rill erosion rate, rill drop height, and rill number, and

its value can be expressed by a multiple nonlinear regression equation [17]. Tang et al. [14] established the relationship between loess slope surface and microtopographic factors, indicating that topographic factors were significantly positively correlated with flow yield and cumulative sediment yield. Luo et al. [18] simulated rainfall on the slope under different tillage methods and pointed out that there was a good quadratic polynomial relationship between surface roughness and sediment yield. The variation characteristics of rill profiles on black soil slope under different slope conditions from the slope surface to the bottom showed "wide and shallow type", "narrow and deep type", and "wide and shallow type" [19]. In addition, microtopography has been identified as a key factor affecting the evolution of soil erosion [20]. It changed the erosivity of runoff and affected the evolution of erosion and sediment loads [21].

The rill development process was complex, and the morphological changes have obvious spatiotemporal evolution characteristics [4,7,22]. With the extension of rainfall duration, the rill network develops from small to large and from simple to complex. Discontinuous rills were gradually connected, and the flow was interconnected successively, that is, the formation of rills can promote runoff concentration and the further development of rills [23,24]. By quantitative measurement, the rill length, width, and depth could be obtained, and some other indicators can be derived to describe the rill shape. Bewket and Sterk [25] directly used the actual rill damage area (i.e., the total rill surface area) to characterize rill erosion. Some scholars used RD to characterize the erosion process [26,27] and believed that RD can better describe the development degree of rill. The higher the RD, the more serious the rill erosion, and this also mean that there were more rill branches. Yan et al. [22] proposed the index of contour deviation degree, rilline surface area, perimeter, diameter, and rilline cutting depth expansion rate to quantitatively describe the morphological characteristics in each stage of rilline development.

With the rapid development of high-tech, high-precision photogrammetry and 3D laser scanning technology, these tools have now been applied to soil erosion research, and the overall morphology of erosion gullies can be extracted by obtaining high-precision DEM [28]. Wang et al. [29] quantified the responses between microtopography and the amount of erosion on overland sand slopes and loess slopes through an indoor artificial simulated rainfall experiment with three-dimensional laser scanning technology. Some scholars also used an ultrasonic distance sensor to explore the water depth detection on the erosion static bed slope for research and development [30], or they used [7]Be tracer technology and artificially simulated rainfall test to study the erosion dynamics of the slope in the process of secondary rainfall [31]. Nouwakpo et al. [32] tested a methodology that inferred the exceedance of transport capacity by concentrated flow from changes to soil surface microtopography sustained during rainfall–runoff events. Qin et al. [33,34] combined with the multicamera stereophotogrammetry technology, compared the high-precision digital elevation model DEM of the slope at different moments, analyzed the morphology of slope rill and the characteristics of rill water flow and their change rules, and verified that the stereophotogrammetry technology could accurately measure real-time dynamic changes of rill morphology.

Although the formation, development, and morphological characteristics of rills have been studied, the mechanism of rill formation and evolution has not been fully understood and mastered. Therefore, it is of great significance to further reveal the mechanism of rill erosion by studying the development process of rill erosion and using corresponding morphological characteristics to describe the morphology of slope rills through a simulated rainfall test method.

## 2. Materials and Methods

### 2.1. Study Area and Experimental Soil

This study was conducted in West Lake Basin, Caidian District, Wuhan City, Hubei Province, which is National Soil and Water Conservation Science and Technology Demonstration Park. The study area is located in the north end of the southern red soil hilly region,

which belongs to the subtropical monsoon climate zone, with abundant precipitation, and the annual average precipitation is 1270 mm. Precipitation varies greatly within the year, mainly concentrated in the wet season from April to September. The average annual temperature ranges from 15.8 °C to 17.5 °C. In this study, the experimental soil was red soil at 0–40 cm surface layer in slope farmland. The mass ratio of organic matter was 14.35 g/kg, and the mass fractions of clay, silt, and sand were 32.6%, 43.9%, and 23.5%, respectively.

The soil samples were retrieved from the field and dried naturally and then screened by 5 mm. Sand 10 cm thick was first filled into the bottom of the trough to facilitate water seepage. Then, according to the soil bulk density of 1.25 g/cm$^3$ measured in the field, soil samples were filled in 4 layers of 10 cm each, and the soil trough was compacted as far as possible around it to prevent the side wall effect. On the day before the rainfall, the rainfall intensity of 30 mm/h was used for 30 min to control the soil moisture content at about 20%, so as to ensure the consistent soil moisture content in the early stage.

### 2.2. Experimental Design

(1) Rainfall test

Rainfall equipment adopted a multinozzle, downjet simulation system. The rainfall intensity varied from 0.5 to 4 mm/min; the rainfall height was 6 m, and the rainfall uniformity was more than 80%. Before the tests, the rain intensity was determined to achieve the designed rain intensity. The rainfall intensity (2.0, 2.5, and 3.0 mm/min) and the slope (6°, 10°, and 15°) were adjusted. Multiple experiments were designed to simulate the sediment yield and runoff process on red soil slope, with two repetitions for each rainfall [35]. The soil trough used in the test was a hydraulic erosion steel trough. The size of the test tank was 3.5 m (length) × 1.0 m (width) × 0.5 m (depth), and the slope adjustment range was 0~20° (Figure 1). The tail of the test soil trough was arranged flush with the soil layer, and the baffle was arranged below. In order to eliminate the boundary effect of soil trough as much as possible, the upper 0.5 m of soil trough was reserved.

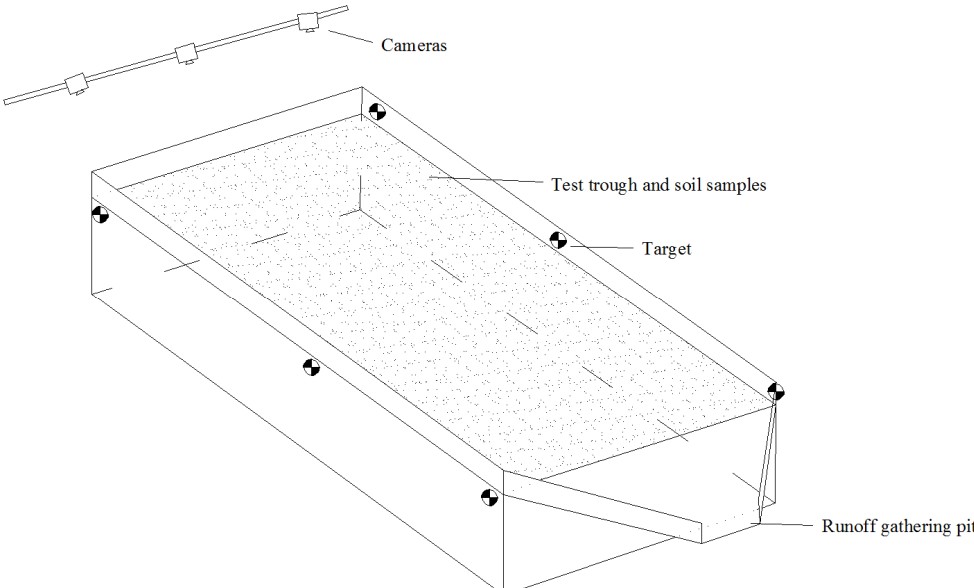

**Figure 1.** Schematic diagram of test device.

We adjusted the slope and then covered the soil trough with a tarpaulin, and rain began. Four rain gauge tubes were uniformly arranged around the soil trough, and the rainfall intensity was determined by receiving rainfall for 3 min. After the rain intensity reached the design requirements, we removed the tarpaulin and started the timer with a stopwatch. Following calibration of the rainfall intensity and uniformity, observations of slope morphology changes were started when rainfall began. Once runoff and sediments

were collected in the sampling buckets, the start time of runoff and sediment yields was recorded, respectively. In the process of rainfall, erosion morphology and rill development on the slope were observed and recorded. After runoff production on the slope, sampling was carried out every 2 min. After the rainfall, runoff and sediment weights were measured by drying and weighing methods, and runoff and sediment yields were calculated.

(2)  Stereophotogrammetric arrangement

Stereoscopic photography technology was used to take the surface morphology photos of the test soil trough at a fixed time, and PhotoScan was used to conduct imaging in the later stage, and the surface microterrain DEM graphics were output to study the morphological characteristics of the rills [33]. The three cameras of the image acquisition system took a single photo of the slope at the same time, and the slope images from three different perspectives were obtained. A series of rill erosion image sets with two dimensions of time and space were obtained by taking continuous photos of slope surface at a certain interval. Structure From Motion (SFM) was the used algorithm to obtain sparse point clouds of rill erosion morphology evolution. The function of the data processing system was to match the same name points in the point cloud, that is, for multiview and high-overlap images, 3D reconstruction and DEM generation were carried out.

Before the test began, three cameras were placed in parallel on the same horizontal line to shoot the soil slope vertically downward. According to the pixel of the cameras (1200 W), we set the vertical distance between the camera and the slope as 1 m, and the adjacent distance between the cameras as 0.3 m. The two camera lenses on both sides were tilted inward by about 5°, and the central camera lens was parallel to the slope. We adjusted the camera's scene mode to "M Manual" so that the aperture was the largest, the sensitivity was the lowest, and the shutter speed was moderate. After the image was clear, we set the focus mode to manual and ensured that the camera's focal length remained unchanged throughout the test. Six black and white targets were set around the test soil trough to keep the target parallel to the soil surface of the test soil trough. Before the test, the relative position of each target was measured with a total station instrument, and a ruler was placed on the slope to ensure the stitching accuracy of the later photos. After capturing multiple sets of photos taken by a multicamera, they were imported into Agisoft PhotoScan Professional software for 3D modeling. Through the three-dimensional model, the coordinate data of the whole slope surface and the elevation data of rill could be obtained. The morphological characteristics of the slope in each period, such as the length, width, and depth of the rills and the spatial distribution of the rills, were analyzed from the DEM map. Combined with the observed data of rill development during rainfall, the morphological characteristics of rill with time could be analyzed.

### 2.3. Calculations and Statistical Analysis

The slope runoff rate is the runoff per unit time and area, which was calculated as follows:

$$R = Q/(AT) \tag{1}$$

where $R$ is the runoff rate, L/(m$^2$ s); $Q$ is the amount of runoff yield, g; $A$ is the soil sample area, m$^2$, and $T$ is rainfall time, min.

The slope sediment yield is the sediment per unit of time and area, which was calculated as follows:

$$E_r = E_a/(AT) \tag{2}$$

where $E_r$ is sediment yield, g/(m$^2$ s), and $E_a$ is drying sediment weight, g.

RD is the length of the rill in a unit study area and represents the distribution of the slope rill, which can be expressed as:

$$\rho = \sum_{j=1}^{m} L_{ij} / A_o \tag{3}$$

where $\rho$ is RD, m/m$^2$; $L_{ij}$ is the total length of rill $j$ on the slope, m, and $m$ is the total number of rills on the slope.

RFD refers to the sum of the total plane area of rills on the slope per unit area. This parameter is a dimensionless parameter, which can reflect the intensity of rill erosion and the degree of slope fragmentation relatively objectively. Its expression was as follows:

$$\mu = \sum_{j=1}^{m} A_j / A_o \tag{4}$$

where $\mu$ is the RFD; $A_j$ is the surface area of rill $j$ on the slope, m$^2$.

RC refers to the ratio of the effective length of the rill to its corresponding vertical effective length, which can reflect the complexity of rill distribution on the slope, and its expression was:

$$c = L_{ij} / L_j \tag{5}$$

where $c$ is RC, and $L_j$ is the vertical effective length of rill $j$ on slope, m.

RWDR refers to the ratio of rill width to its corresponding depth, which can reflect the morphological characteristics of rill grooves on the slope, and its expression could be expressed as follows:

$$R_{WD} = \sum_{i=1}^{n} W_i / \sum_{i=1}^{n} D_i \tag{6}$$

where $R_{WD}$ is the RWDR; $W_i$ is the rill width measured at the $i$th point on the slope, m, and $D_i$ is the rill depth measured at the $i$th point on the slope, m.

PhotoScan software was used to reconstruct the rill morphological 3D model. Origin 9 was used for mathematical statistical analysis and plotting of the test data. SPSS 19.0 was used for variance analysis of the combined test. The minimum significant difference (LSD) method was used for multiple comparisons of the test results. The dependent variables were tested for difference significance under different conditions (Mann–Whitney U test and paired sample nonparametric test, $\alpha = 0.05$).

## 3. Results

### 3.1. Process of Runoff and Sediment Yield on Slope

With the increase in rainfall duration, the runoff yield on the slope gradually increased and tended to be stable under different rainfall intensities and slopes (Figure 2). When the slope was 15°, the greater the rainfall intensity, the greater the peak flow yield. When the rainfall intensity was 3.0, 2.5, and 2.0 mm/min, the peak yields reached 2.50, 1.97, and 1.84 L (m$^2$ s)$^{-1}$. This was mainly because under the same slope infiltration rate, the greater the rainfall intensity, the greater the conversion of precipitation to surface runoff. When the rainfall intensity was 3.0 mm/min, the variation range of runoff yield increased with the increase in slope after reaching dynamic stability. The impact of slope change on flow yield increases and decreases, but with the increase in slope, the peak value of flow yields increased roughly. At slopes of 6°, 10°, and 15°, the peak flow yields reached 1.77, 1.85, and 2.50 L (m$^2$ s)$^{-1}$.

Under the same slope condition, the erosional sediment yield on the slope increased sharply with the increase in rainfall intensity at first and then decreased gradually to the peak value until it reached dynamic stability (Figure 3a). The peak value of sediment yield under 3.0 mm/min rainfall intensity (0.74 g (m$^2$ s)$^{-1}$) was 1.34 times and 2.96 times that under 2.5 and 2.0 mm/min rainfall intensity. With the same rainfall intensity (3.0 mm/min), the sediment yield on the slope was positively proportional to the slope (Figure 3b). The peak value of sediment yield reached about 10 min after rainfall, and the peak value of sediment yield increased significantly with the increase in slope. The peak sediment yield (1.44 g (m$^2$ s)$^{-1}$) on the 15° slope was 1.95 times and 11.74 times that on the 10° and 6° slopes. In the process of early rainfall, there was no rill erosion, only splash erosion. The runoff form was a slope thin-layer flow; its kinetic energy was small, and the sediment yield

process was mainly the transport of spattered soil particles and the layered denudation of the soil surface under inter-rill erosion. With the increase in rainfall duration, surface particles were transported, runoff intensity increased, and the soil crust was destroyed. The sediment yield increased significantly after the rill appeared but decreased and tended to be stable after the dynamic equilibrium.

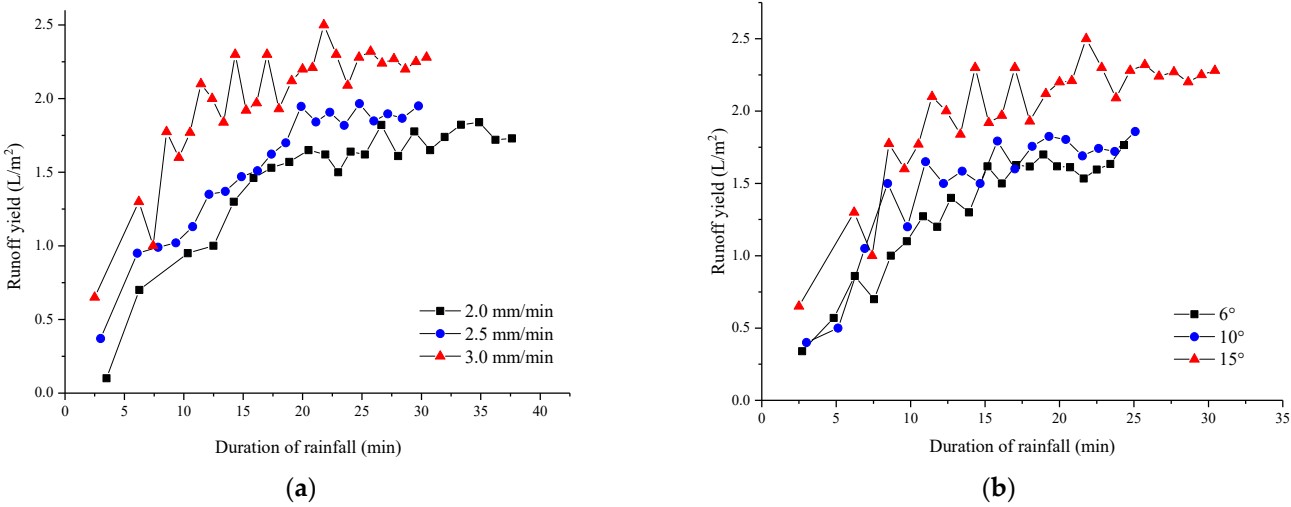

**Figure 2.** Runoff yield process of slope. (**a**) Slope was 15°.and (**b**) rainfall intensity was 3.0 mm/min.

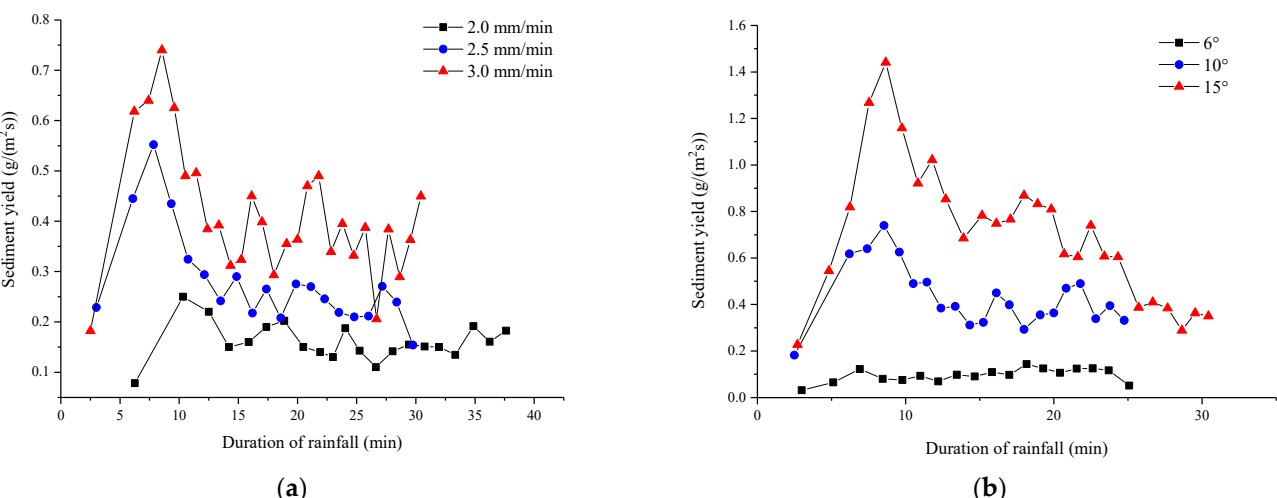

**Figure 3.** Sediment yield process of slope. (**a**) Slope was 15° and (**b**) rainfall intensity was 3.0 mm/min.

### 3.2. Variation of Rill Morphological

On the 15° slope, when the rain intensity was low (2.0 mm/min), rill development was slow, and the change was not obvious (Figure 4a). When the rainfall intensity increased to 2.5 mm/min, 3 to 5 scattered rills appeared after 5 min of rainfall; the longest rill was up to 1.2 m, the width was 1.83 cm, and the depth was up to 0.6 cm (Figure 4b). With the increase in rainfall duration, the length, width, and number of rills increased significantly. After 20 min of rainfall, the shape of the rills did not change significantly, and the maximum length was 2 m; it was 3.7 cm wide and 1.3 cm deep. When the rainfall intensity increased to 3.0 mm/min, rill development was obvious after 5 min of rainfall, and there was an obvious main rill, and the number of rills (more than 7) and length, width and depth (up to 1.47 m, 2.85 cm, and 0.9 cm, respectively) were all larger than that when the rainfall intensity was 2.5 mm/min (Figure 4c). Moreover, rill morphology changed significantly with the increase in rainfall duration. In particular, after 30 min, the rill developed into two

distinct main rills with a length of 2.58 m and a maximum width of 9.7 cm and a maximum depth of 2.2 cm.

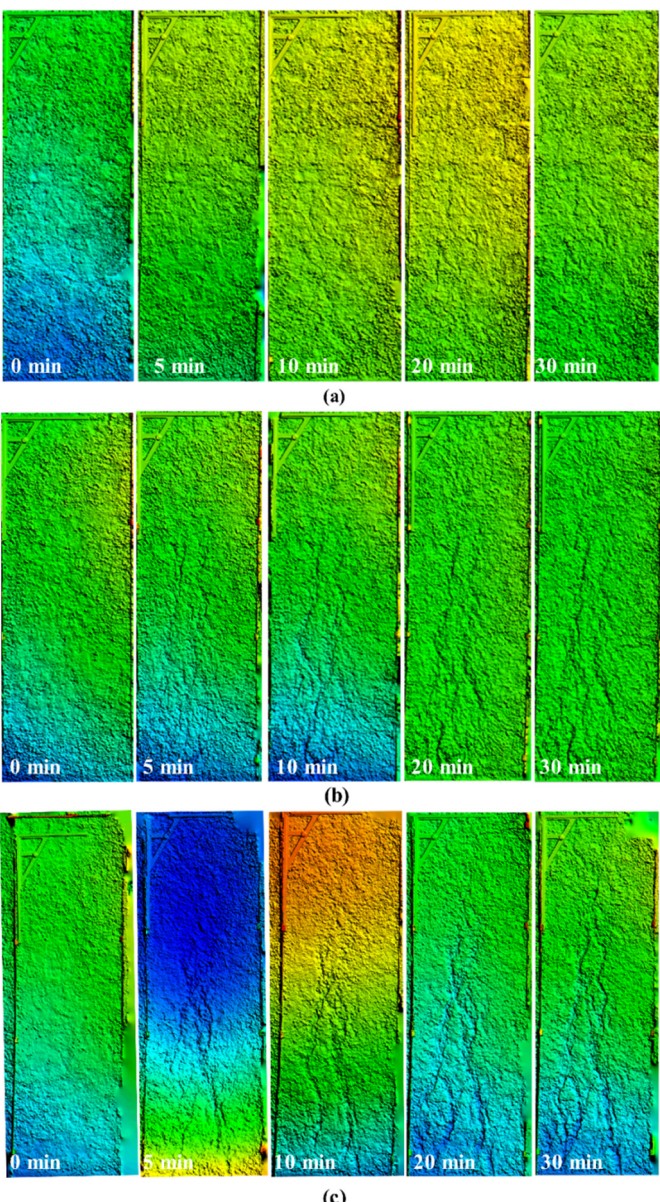

**Figure 4.** DEM map at fixed time under different rainfall intensities. (**a**) rainfall intensity was 2.0 mm/min; (**b**) rainfall intensity was 2.5 mm/min and (**c**) rainfall intensity was 3.0 mm/min.

The rill development morphology was significantly different under different slopes than under different rain intensities (Figure 5). Under the same rain intensity (3.0 mm/min), when the slope was 6°, a single shallow rill appeared 5 min after the rainfall, with the maximum length, width, and depth of 1.30 m, 5.6 cm, and 0.11 cm, respectively (Figure 5a). With the increase in rainfall duration, the rill tended to be traced and widened, but the change was not obvious, and no obvious branch appeared. When the slope was 10°, small and disorderly gullies appeared on the slope after 5 min of rainfall, with the maximum length, width, and depth of 1.07 m, 4.0 cm, and 0.14 cm, respectively (Figure 5b). With the increase in rainfall duration, the number of rills and length and width of the rills gradually increased. After 30 min, the maximum length and width of the rills increased by 1.05 to 1.36 times, but no main rill developed. When the slope was 15°, multiple rills appeared after 5 min of rainfall, and the rills developed significantly with increasing duration, and

the number and length of the rills increased continuously (Figure 5c). In general, rainfall intensity has a greater impact on rill development than slope.

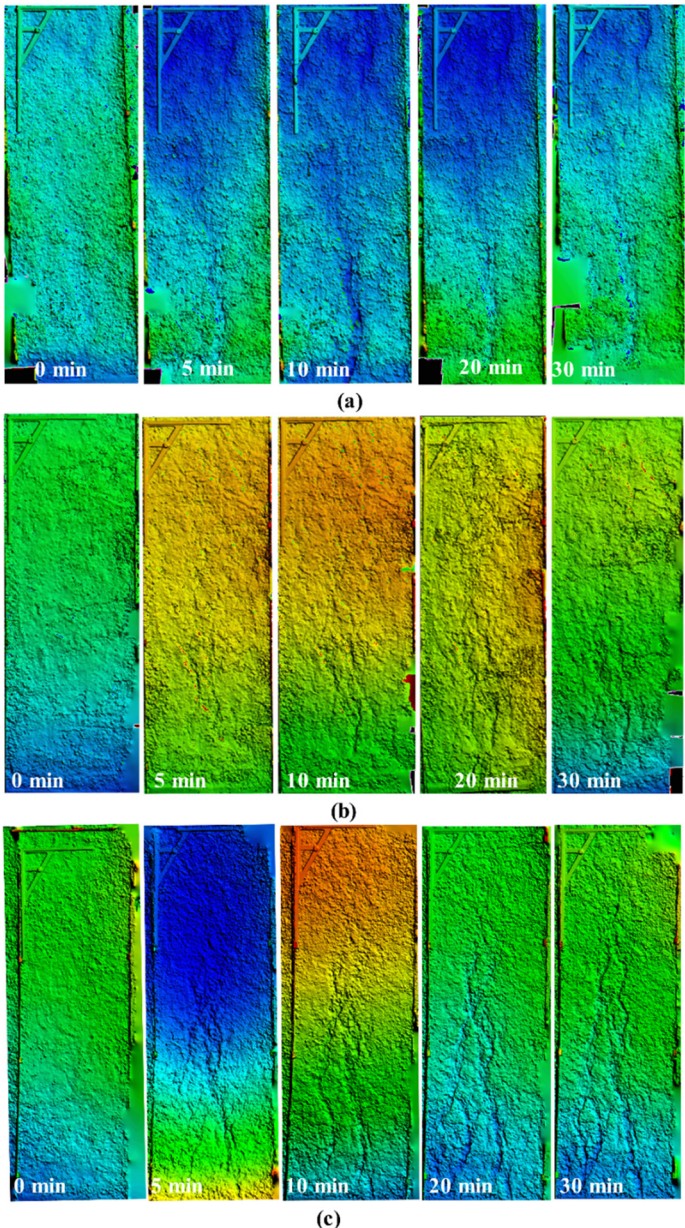

**Figure 5.** DEM map at fixed time under different slopes. (**a**) slope was 6°; (**b**) slope was 10° and (**c**) slope was 15°.

### 3.3. Variation of Rill Morphological Indexes

Based on the rill morphology, the PhotoScan software was used to reconstruct the rill morphological 3D model to extract and calculate the indicators of RD, RFD, RC, and RWDR. The four indicators were selected to quantify rill erosion and rill morphology from different perspectives after rainfall. RD, RFD, and RC all increased with the increase in rainfall intensity and slope, but the RWDR changed on the contrary (Figure 6). When the rainfall intensity was 3.0 mm/min, the slope increased from 6° to 10° and 15°; RD increased by 16.67% and 40.63%; RFD increased by 16.67% and 33.33%, and RC increased by 10.62% and 23.89%. In addition, RWDR decreased by 35.50% and 56.28%. When the slope was 15°, rainfall intensity increased from 2.0 mm/min to 2.5 and 3.0 mm/min; RD increased by 21.69% and 62.65%; RF increased by 50.00% and 100.00%, and RC increased by 13.08% and 30.84%. In addition, RWDR decreased by 15.82% and 36.08%. The greater the rainfall

intensity and slope, the richer the rill network, that is, the greater the rill complexity. The decrease in RWDR indicated that the increase in rill erosion intensity was greater than that of rill wall collapse. Rain intensity had different effects on the four indexes, among which, RD, RC, and RFD were more sensitive (with a large variation range). However, RWDR was more sensitive to the increase in slope.

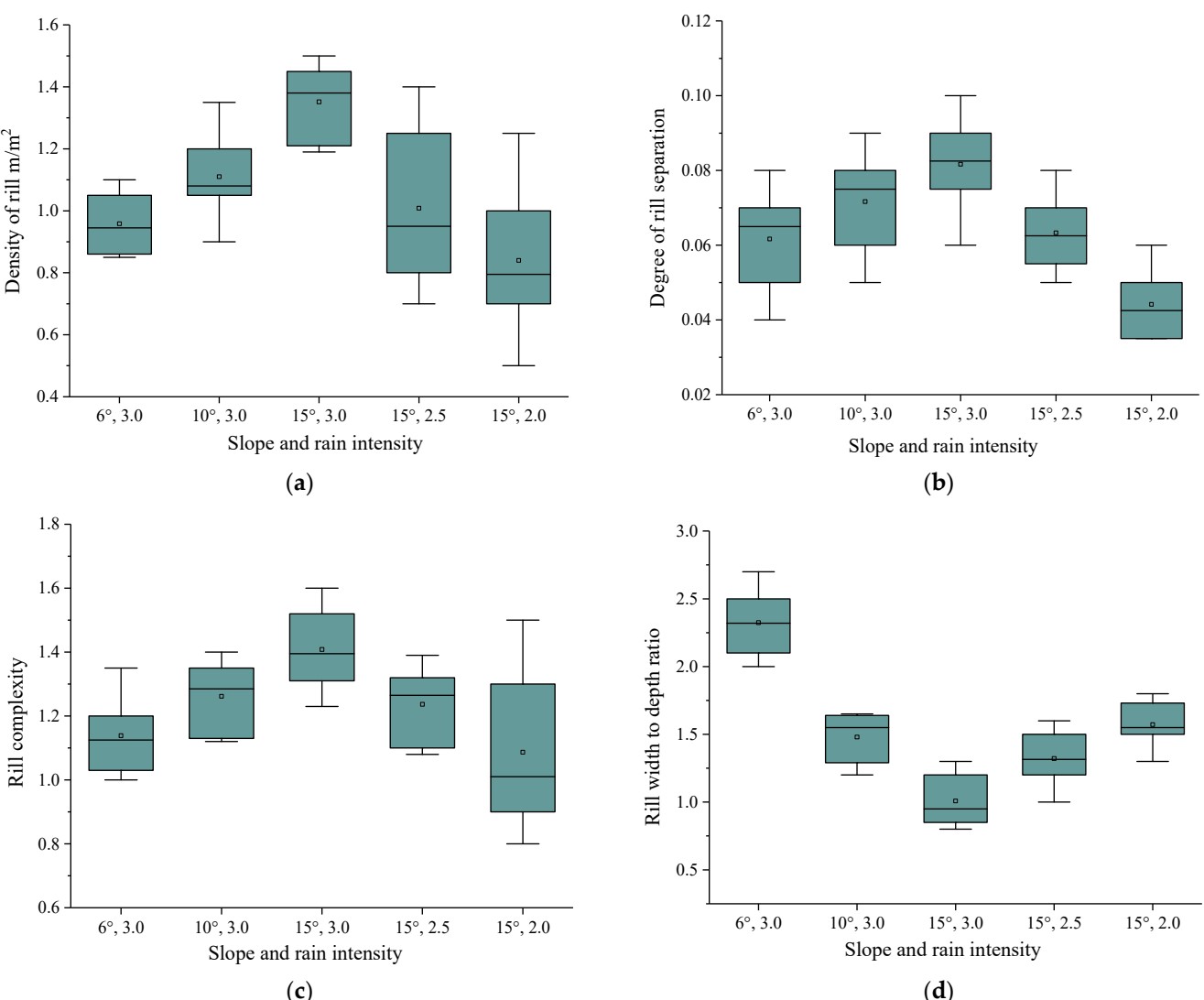

**Figure 6.** Variation of rill morphological indexes under different slopes and rainfall intensities. (**a**) density of rill; (**b**) degree of rill separation; (**c**) rill complexity and (**d**) rill width to depth ratio.

*3.4. Simulation of Sediment Yield on Rill Slope*

There was a significant positive correlation between the rill erosion rate (RER) and RD, RC, and RFD and a significant negative correlation between RWDR (Table 1). Of the correlations among different morphological indexes, RC had the highest correlation with the other three indexes (0.85). Therefore, RC can be regarded as the best index, and the other indexes can be analyzed as related indexes.

**Table 1.** Correlation analysis between rill erosion rate and four rill morphology indexes.

| | RER kg/(m$^2$·s) | RD (m·m$^{-2}$) | RFD | RC | RWDR |
|---|---|---|---|---|---|
| RER | 1 | | | | |
| RD | 0.75 | 1 | | | |
| RFD | 0.76 | 0.91 | 1 | | |
| RC | 0.85 | 0.82 | 0.89 | 1 | |
| RWDR | −0.73 | −0.84 | −0.77 | −0.90 | 1 |

The regression analysis of slope sediment yield with RD, RC, RFD, and RWDR showed that the relationship between sediment yield and the rill morphological indexes was a power function (Y = aX$^b$), but the correlation coefficients were low (R$^2$ < 0.8). The sediment yield was selected to perform multiple nonlinear regressions fitting with the above indicators, and it was found that the sediment yield had a good nonlinear relationship with RC and RWDR:

$$E_r = 0.17\,c^{\,6.75}R_{WD}{}^{1.30} \ (\text{R}^2 = 0.89,\ \text{NSE} = 0.85,\ \text{n} = 10)$$

RC can reflect the richness of the slope rill network. The rill network tended to be more complex and richer, and the low rainfall intensity was more beneficial to the development and perfection of the slope rill network. The RC index can represent the expansion degree of the rill network and was a good index for measuring rill morphology on the slope. It can also reflect the bifurcation, merging, and connectivity of rills on the slope. RWDR can objectively reflect the degree of slope fragmentation and rill erosion intensity.

## 4. Discussions

On the path of slope flow, the erosive force increased gradually until it was able to detach and transport soil blocks, and then erosion took shape [4]. The slope step–pools scoured by runoff on the slope were a symbol of the beginning of rill erosion. The connection between the step–pools means the development of rill erosion. The formations of the step–pools were related to the weak antiscrunch property of the local slope. The soil structure on the natural slope was usually uneven, and the weak soil antiscourability will first produce strong local erosion and form the step–pools [36]. During rill development, rill wall collapse erosion and rill retrogression erosion occurred simultaneously. The collapse of the ditch wall was due to the erosion and transport of the runoff in the ditch, which caused the ditch wall to expand outward gradually with the development of the rill [37]. Rill headward erosion was the upward development of the source due to the shear force of the runoff in the rill [38]. In this study, the greater the rainfall intensity, the more runoff in the rill, that is, the stronger the erosion capacity. The increase in slope accelerated rill development and rill erosion, and rill formation will affect the sediment yield on the slope. Combined with the special time nodes of rill development, it can be found that after the initial sharp increase in runoff yield, the slope formed a stable streamflow and randomly distributed step–pools. After the formation of the step–pools, the flow rate curve was relatively smooth, but it was still in a steady growth stage until the step–pools were connected (Figures 4 and 5). With the increase in slope and rain intensity, the formation and connection time of the rills will advance, and the arrival time of peak flow and sediment yield will shorten (Table 2). At the early stage of sediment yield, the amount of erosion increased at a uniform rate, but after rill erosion began to develop, the amount of erosion increased sharply. The peak value of sediment yield on the slope was very similar to the occurrence of the time of the connection of step–pools. It is said that the development of rill erosion led to the rapid increase in sediment yield on the slope (Table 2).

**Table 2.** Time nodes of rill development and sediment yield rate.

| Slope | Rainfall Intensity | Formation Time of Step–Pools | Connection Time of Step–Pools | Peak Sediment Yield (g/s) | Peak Last |
|---|---|---|---|---|---|
| 6° | 3.0 | 3′00 | 8′20 | 7.97 | 7′35 |
| 10° | 3.0 | 3′40 | 5′40 | 12.72 | 7′33 |
| 15° | 3.0 | 3′03 | 4′30 | 12.30 | 4′21 |
| 15° | 2.5 | 3′10 | 7′17 | 8.59 | 7′20 |
| 15° | 2.0 | 3′00 | 10′33 | 2.39 | 9′40 |

Stereophotogrammetry technology extracted high-precision DEM based on the overlapping parts of different photos taken by digital cameras at the same time [33,39]. Different cameras can realize real-time accurate measurements of slope microtopography and conduct a three-dimensional analysis of the surface morphology of the measured object [40,41]. Based on stereoscopic photogrammetry, the variation of runoff and sediment yields on the slope surface in this study was closely related to erosion morphology, and the rill morphological parameters can be extracted for further analysis. RC directly reflected the richness of the rill network and the expansion degree of the rill network. It can also reflect the bifurcation, merging, and connectivity of rills on the slope [22]. RWDR objectively reflected the change of rill and groove shapes under different conditions [42]. In this study, RC ranged from 0.86 to 1.40, and RWDR ranged from 1.01 to 2.79. RC and RWDR, as morphological indicators, could be used to characterize erosion intensity, which is different from other studies [4]. With the increase in rainfall intensity and slope, the average flow velocity of the rill current also showed an increasing trend. The increase in flow velocity will increase the erosion intensity of upstream traceability and reduce the runoff confluence time at the initial stage of overland runoff production. Then, the erosion of the rill edge wall by the current rill gradually intensified and resulted in the final increase in the rill area. Therefore, RD, RC, and RFD naturally increased. Due to rill development, the increase in headward erosion intensity was greater than that of the gully wall collapse. With the increase in slope, the stress state of runoff and surface soil on the slope were also changed. The greater the slope, the more enhanced the scouring ability of the current in the rill to the soil groove at the bottom of the rill, resulting in a further increase in rill depth and a further decrease in RWDR.

In the process of erosion, splash erosion was the main erosion at the start of rainfall, and then surface erosion along with the slope surface thin-layer runoff developed. When the runoff constantly collected and the erosion energy was large enough, rill erosion occurred on the slope surface, and the shallow gully and cut gully gradually developed downslope if the slope was long enough. At present, the research focus of erosion models based on physical processes was still on slope and small watershed scales, while the spatial scale differences involved in large-scale watershed were more complex. InVEST (Integrated Valuation of Environmental Services and Trade-offs) is a tool that incorporates the process of erosion and sedimentation of watersheds. It can be applied to watershed and subwatersheds and gives an indication of the destination of eroded particles using the Sediment Delivery Ratio (SDR). This model can be calibrated by comparing the inputs of the estimated sediments exported with the rate of siltation in the reservoirs [43]. SWAT (Soil and Water Assessment Tool) is a distributed model at the watershed scale. It was used to model and predict the effects of long-term land management practices on runoff, sediment loads, and nutrient losses in a large and complex watershed with a variety of soil types, land uses, and management conditions [44]. The distributed model divided the watershed into several grids or representative basic units and reflected the differences of various factors affecting soil erosion in the watershed by assigning values to the calculation units. Then, according to a series of operation programs reflecting the erosion process, the runoff and sediment yields of the unit were calculated, so as to achieve the purpose of accurately predicting the whole basin. However, it was difficult to convert the scale from slope to watershed, which involved the transport and deposition of runoff and sediment.

Whether and how small-scale process models can be integrated into large-scale study areas still needs to be solved. It was an effective way to realize watershed scale prediction to observe multiscale erosion processes by using high-precision measurement methods. In this study, the relationship between slope erosion and rill morphology was quantified by photogrammetry technology, which can provide a reference for erosion intensity estimation at watershed scale.

## 5. Conclusions

In order to quantitatively study the characteristics and influencing mechanisms of rill morphology development during slope erosion, a series of simulated rainfall experiments were carried out by using stereophotogrammetry. The sediment yield on the slope was closely related to rill morphology change during the rainfall. With the increase in rainfall duration, the variation of runoff yield and sediment yield on the slope was different, but both tended to be stable after reaching the threshold value. The length, width, and number of rills were positively correlated with rainfall intensity and slope. RD, RFD, RC, and RWDR can be used as the best indicators to characterize rill morphology. The relationship between sediment yield and RC and RWDR was nonlinear.

**Author Contributions:** Conceptualization, Z.Z. and H.W.; methodology, Z.Z. and H.W.; validation, Z.Z. and S.D.; formal analysis, Z.Z., H.W., S.D. and Y.W.; investigation, Z.Z., H.W., S.D. and Y.W.; resources, Z.Z. and H.W.; data curation, Z.Z., H.W. and Y.W.; visualization, Z.Z., H.W., S.D. and Y.W.; writing—original draft preparation, Z.Z., H.W. and Y.W.; writing—review and editing, Z.Z., H.W., S.D. and Y.W.; project administration, H.W.; funding acquisition, S.D. All authors have read and agreed to the published version of the manuscript.

**Funding:** This research was funded by the Program of National Key Research and Development of China (2016YFC0503506) and the "Soil and Water Conservation Monitoring Research in Hubei Province" project (2017052601) funded by Hubei Anyuan Safety and Environmental Protection Technology Co., Ltd.

**Acknowledgments:** All authors thank the editors and reviewers for insightful comments on the original manuscript.

**Conflicts of Interest:** The authors declare no conflict of interest.

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
