# Peer review of "Effect of Rill Development on Slope Erosion and Sediment Yield Based on Stereophotogrammetry Technology"

_water, doi:10.3390/w14192951_

Round 1

Reviewer 1 Report

The manuscript (MS) is devoted to assessment of the effect of rill development on slope erosion and sediment yield based on stereo photogrammetry technology. The results of this study are new and can be potentially of interest to the readership of Water. However, I cannot recommend a publication of the MS in its present form. I am inclined to propose a major revision.

My concerns and suggestions are as follows.

(1) Abstract: It is suggested to supplement the research background, the significance of the research results, etc., and reorganize the sentences according to the sequence number.

(2) Introduction needs to be concentrated and focus the topic. Please delete general knowledge.

L36-37: Whether rill erosion was studied?

L56: Where are the shortcomings or deficiencies of the existing research, and where can this research be improved?

L65-72: Please make objectives concise and clear.

L70-71: Sloping farmland is not mentioned in the entire foreword.

(3) The introduction of materials and methods is not clear enough, and relevant pictures need to be added. Lack of design rationale? Slope, slope length, etc.

L114-140: Recheck the equation.

(4) Results: This section is more descriptive. It needs to be revised. The results section needs to indicate the representativeness of the selected current experimental conditions.

L146-148: Runoff and infiltration should be the reverse law.

L150: Is the data reversed, the higher the slope, the higher the runoff rate?

L169: How are these figures calculated.

Part 3.3 Should be the more important content of the full text, it is suggested to describe in detail.

(5) The discussion chapter needs to be revised. Focus on the problems found in the results of the experiment rather than the plain narrative. It is recommended to supplement relevant diagrams for in-depth exploration.

(6) In general, it is possible to understand the MS text, but it would substantially benefit from a professional English editing, in particularly the style. Especially often, there is a confusion of adverbs and adjectives (i.e. "significant greater than" instead of "significantly greater than").

Author Response

Reviewer 1

The manuscript (MS) is devoted to assessment of the effect of rill development on slope erosion and sediment yield based on stereo photogrammetry technology. The results of this study are new and can be potentially of interest to the readership of Water. However, I cannot recommend a publication of the MS in its present form. I am inclined to propose a major revision.

My concerns and suggestions are as follows.

(1) Abstract: It is suggested to supplement the research background, the significance of the research results, etc., and reorganize the sentences according to the sequence number.

Thanks for your insightful comment. Abstract has been revised. Please see line 11, 20-22.

(2) Introduction needs to be concentrated and focus the topic. Please delete general knowledge.

Sorry for that confusion. Introduction has been revised.

L36-37: Whether rill erosion was studied?

Please see line 34-38.

L56: Where are the shortcomings or deficiencies of the existing research, and where can this research be improved?

Please see line 68-71.

L65-72: Please make objectives concise and clear.

Please see line 68-71.

L70-71: Sloping farmland is not mentioned in the entire foreword.

It has been deleted.

(3) The introduction of materials and methods is not clear enough, and relevant pictures need to be added. Lack of design rationale? Slope, slope length, etc.

L114-140: Recheck the equation.

Thanks for your insightful comment. The equation has been checked and revised. Please see line 139-153.

(4) Results: This section is more descriptive. It needs to be revised. The results section needs to indicate the representativeness of the selected current experimental conditions.

Thanks for your insightful comment. Results have been revised.

L146-148: Runoff and infiltration should be the reverse law.

Please see line 162-164.

L150: Is the data reversed, the higher the slope, the higher the runoff rate?

Sorry for that confusion. It has been revised. Please see line 166-167.

L169: How are these figures calculated.

Please see line 137-153.

Part 3.3 Should be the more important content of the full text, it is suggested to describe in detail.

Thanks for your insightful comment. More details have been added. Please see line 211-221.

(5) The discussion chapter needs to be revised. Focus on the problems found in the results of the experiment rather than the plain narrative. It is recommended to supplement relevant diagrams for in-depth exploration.

Thanks for your insightful comment. Please see line 248-253, 267-293.

(6) In general, it is possible to understand the MS text, but it would substantially benefit from a professional English editing, in particularly the style. Especially often, there is a confusion of adverbs and adjectives (i.e. "significant greater than" instead of "significantly greater than").

Thanks for your comment. It has been revised.

Reviewer 2 Report

This manuscript is aiming to quantify the effect of rill development on slope erosion and sediment yield. The topic was interesting. The present manuscript was well organized and written. The experimental methods and research contents are innovative. This study has some interesting in the quantitative study of rill development and mechanism of soil erosion. I suggest the manuscript can be accepted for publication in the journal after minor revision has been made. Following issues still should be addressed by authors.

 1. The logicality of the content of "Introduction" should be revised carefully.

2. There are some details errors in this manuscript. Such as line 10-12, 254. Carefully review the full manuscript.

3. Pictures of the experimental setup are missing. Supplement the experiment photos to help readers better understand the experiment process and method.

4. How to consider the designed rain intensity and slope in the part of "2.1 Experimental materials and equipment "?

5. How to determine the start time of runoff and sediment yield? Will they be determined on the basis of experimental phenomena merely?

6. In Fig.1 and Fig.2, production flow rate and sediment yield rate varying with duration of rainfall (time) presents different change laws, the reasons should be highlighted and supplemented in detail.

7. Discussion: It needs to be revised. The reasons and mechanism of rill development and runoff and sediment yields should be explained in detail.

Author Response

Reviewer 2

Comments and Suggestions for Authors

This manuscript is aiming to quantify the effect of rill development on slope erosion and sediment yield. The topic was interesting. The present manuscript was well organized and written. The experimental methods and research contents are innovative. This study has some interesting in the quantitative study of rill development and mechanism of soil erosion. I suggest the manuscript can be accepted for publication in the journal after minor revision has been made. Following issues still should be addressed by authors.

  1. The logicality of the content of "Introduction" should be revised carefully.

Thanks for your insightful comment. The content of "Introduction" has be revised carefully. Please see line 34-43.

  1. There are some details errors in this manuscript. Such as line 10-12, 254. Carefully review the full manuscript.

Sorry for that confusion. The full text has been checked and detailed errors have been corrected.

  1. Pictures of the experimental setup are missing. Supplement the experiment photos to help readers better understand the experiment process and method.

Thanks for your insightful comment. Figures of the experimental setup have been added.

  1. How to consider the designed rain intensity and slope in the part of "2.1 Experimental materials and equipment "?

Thanks to the comment. The rainfall intensity was mainly based on the single time rainfall intensity observed by the field test station on the study area. The rainfall intensity for similar studies was also referred. We have considered the factor in the experiment and provided reference.

  1. How to determine the start time of runoff and sediment yield? Will they be determined on the basis of experimental phenomena merely?

Thank you very much for the comment. Runoff and sediments on the slopes were collected in plastic buckets through funnel-shaped collecting troughs in this study. Once the slope begins to produce flow or sediments, the plastic buckets collect runoff or sediments that have been carried by erosion power. Generally, due to the transport of runoff and sediments, the start time of runoff yield lags behind the start time of rainfall, and sediment yield lags behind the start time of runoff yield. Therefore, the beginning time of sand production was measured, and the slope experimental phenomenon was just to help us better judge. We have provided more information about the experimental methods. Please see line 100-102.

  1. In Fig.1 and Fig.2, production flow rate and sediment yield rate varying with duration of rainfall (time) presents different change laws, the reasons should be highlighted and supplemented in detail.

Thanks for your insightful comment. Please see line 162-164, 177-181.

  1. Discussion: It needs to be revised. The reasons and mechanism of rill development and runoff and sediment yields should be explained in detail.

Thanks for your insightful comment. Please see line 248-253, 267-293.

Reviewer 3 Report

Dear authors, I must congratulate on your rigorous experimental study of assessing the impact of rill development on the soil erosion. I have certain queries that need to be clarified in the manuscript.

1. What area of the study area studied?

2. Photogrametry related details are not provided much in the manuscript.

3. How can the results from this study be integrated in sediment yield assessment models such as SWAT, INVEST SDR

4. This seems to be a very localized study. Usually, rill development occurs in watersheds, and we need more elaborate and detailed information regarding the watershed processes. How can this study help?

These issues need to be clarified in the manuscript itself.

Author Response

Reviewer 3

Comments and Suggestions for Authors

Dear authors, I must congratulate on your rigorous experimental study of assessing the impact of rill development on the soil erosion. I have certain queries that need to be clarified in the manuscript.

  1. What area of the study area studied?

Thanks for your insightful comment. The study area profile has been added in the manuscript. Please see Line 73-85.

  1. Photogrametry related details are not provided much in the manuscript.

Sorry for that confusion. We have added information about photogrammetric methods.

Please see Line 107-115.

  1. How can the results from this study be integrated in sediment yield assessment models such as SWAT, INVEST SDR

Thanks for your insightful comment. InVEST is a tool that incorporates the process of erosion and sedimentation of watersheds. It can be applied to watershed and subwatersheds and gives an indication of the destination of eroded particles using the Sediment Delivery Ratio (SDR). This model can be calibrated by comparing the inputs of the estimated sediments exported with the rateof siltation in the reservoirs. SWAT (Soil and Water Assessment Tool) is a distributed model at the watershed scale. It was used to model and predict the effects of long-term land management practices on runoff, sediment loads, and nutrient losses in a large and complex watershed with a variety of soil types, land uses, and management conditions. However, it was difficult to convert the scale from slope to watershed, which involved the transport and deposition of runoff and sediment. Whether and how small-scale process models can be integrated into large-scale study areas still needs to be solved. It was an effective way to realize watershed scale prediction to observe multi-scale erosion process by using high-precision measurement methods. Please see Line 275-293.

  1. This seems to be a very localized study. Usually, rill development occurs in watersheds, and we need more elaborate and detailed information regarding the watershed processes. How can this study help?

Thanks for your insightful comment. In the process of erosion, spash erosion was the main erosion at the beginning rainfall and then the surface erosion along with the slope surface thin layer runoff development. When the runoff constantly collects and the erosion energy was large enough, rill erosion occured on the slope surface, and the shallow gully and cut gully were gradually developed downslope if the slope was long enough. At present, the research focus of erosion models based on physical processes was still on slope and small watershed scales, while the spatial scale differences involved in large-scale watershed were more complex. In this study, the relationship between slope erosion and rill morphology was quantified by photogrammetry technology, which can provide reference for erosion intensity estimation at watershed scale. Please see Line 275-293.

These issues need to be clarified in the manuscript itself.

Thanks for your insightful comment. The above issues have been modified in the article.

Round 2

Reviewer 1 Report

The author has made improvements to the revision comments and can be accepted

Reviewer 3 Report

Thank you very much for considering my comments.